# The Difference of Cholesterol, Platelet and Cortisol Levels in Patients Diagnosed with Chronic Heart Failure with Reduced Ejection Fraction Groups According to Neutrophil Count

**DOI:** 10.3390/medicina57060557

**Published:** 2021-06-01

**Authors:** Aušra Mongirdienė, Jolanta Laukaitienė, Vilius Skipskis, Lolita Kuršvietienė, Julius Liobikas

**Affiliations:** 1Department of Biochemistry, Medicine Academy, Lithuanian University of Health Sciences, Eiveniu Str. 4, LT-50103 Kaunas, Lithuania; jolanta_laukaitiene@yahoo.co.uk (J.L.); lolita.kursvietiene@lsmuni.lt (L.K.); julius.liobikas@lsmuni.lt (J.L.); 2Cardiology Clinic, University Hospital, Lithuanian University of Health Sciences, Eiveniu Str. 2, LT-50161 Kaunas, Lithuania; 3Laboratory of Molecular Cardiology, Institute of Cardiology, Lithuanian University of Health Sciences, Sukilėlių pr.15, LT-50103 Kaunas, Lithuania; skipskis@gmail.com; 4Laboratory of Biochemistry, Neuroscience Institute, Lithuanian University of Health Sciences, LT-50161 Kaunas, Lithuania

**Keywords:** platelet, neutrophil, monocyte, lymphocyte, cortisol, heart failure, inflammation, stress, thrombosis, lipidaemia

## Abstract

*Background and Objectives*: It is known that neutrophils are involved in the pro-inflammatory processes and thus, can have a great impact on the pathophysiology of heart failure (HF). Moreover, hypercholesterolemia heightens neutrophil production, thereby accelerating cardiovascular inflammation. However, there is a lack of information about the relation of low inflammation to the state of stress, hypercholesterolemia, and pro-thrombotic statement in patients with chronic HF. Therefore, we aimed to determine whether platelet, cholesterol and cortisol levels differ in a different inflammatory condition groups according to the neutrophil count in patients diagnosed with CHF with reduced ejection fraction (CHFrEF), and whether there is a correlation between those readings. *Materials and Methods:* The average of neutrophil count was 4.37 × 10^9^ L; therefore, 180 patients were separated into two groups: one with relatively a higher inflammatory environment (neutrophil count ≥ 4.37 × 10^9^ L (*n* = 97)) and one with a relatively lower inflammatory environment (neutrophil count < 4.38 × 10^9^ L (*n* = 83)). We also determined the levels of lymphocytes, monocytes, platelet count (PLT), mean platelet volume (MPV), platelet aggregation, the levels of cortisol and cholesterol and the concentrations of C reactive protein (CRP) and fibrinogen. *Results:* We found that CRP, fibrinogen and cortisol concentrations were statistically significantly higher in the group with higher neutrophil counts. However, there were no differences among cholesterol concentration and other markers of platelet function between the groups. We also showed that PLT, leukocyte and monocyte counts were higher in the group with a higher neutrophil count, and the PLT correlated with other cell type count and CRP. In addition, the neutrophil count correlated with concentrations of fibrinogen, evening cortisol and CRP. *Conclusions:* Cortisol, fibrinogen and CRP levels, PLT and monocyte counts were higher in the CHFrEF patient group with higher neutrophil counts. The cholesterol levels and platelet function readings did not differ between the groups. The neutrophil count correlated with evening cortisol concentration.

## 1. Introduction

It is well documented that chronic inflammation can lead to cardiovascular diseases (CVD) [1,2]. In addition, the importance of neutrophils for the development of cardiovascular inflammation has also been demonstrated [3]. Thus, under hypercholesterolaemia, the production of neutrophils in the bone marrow and at extramedullary sites becomes intensified, thereby accelerating the development of cardiovascular inflammation. Moreover, certain lifestyle factors, including stress, can promote the cardiovascular inflammation through neutrophil production as well. Neutrophil-driven macrophage activation and stimulation of coagulation both observed under cardiac hypertrophy and stroke conditions also have a negative effect [3]. It is worth nothing that the count of neutrophils close to the upper limit of the standard range has been found to correlate with the increased probability of arterial thrombosis in patients with heart failure (HF) [4]. Such a complication could be associated with interaction between neutrophils and the platelets or macrophages, and thus it could intensify the neutrophil-dependent inflammatory reactions [5,6]. It is likely that neutrophils can participate in the pro-inflammatory processes and thus, can have a great impact on the pathophysiology of both the reduced and preserved chronic HF (CHF) [7]. Moreover, it is also known that most of all CHF cases with reduced ejection fraction (CHFrEF) can lead to the ischemic heart disease whose development depends on hypercholesterolemia. However, there is a lack of information regarding the association of cholesterol and a stress marker cortisol with an inflammatory condition in the progression of CHF. Thus, our aim was to determine whether platelet, cholesterol and cortisol levels differ in different inflammatory condition groups according to the neutrophil count in patients diagnosed with CHF with reduced ejection fraction (CHFrEF), and whether there is a correlation between those readings. We suggest that our findings would allow to better understand the differences between cholesterol concentration, low inflammation and stressful conditions in CHFrEF patients with different pro-inflammatory statement, and to find new possible treatment targets and directions for future investigation.

## 2. Methods

### 2.1. Study Population

A total of 356 patients admitted to the Department of Cardiology, of Kaunas Clinical Hospital of Lithuanian University of Health Sciences between 1 January 2016–1 March 2018 and diagnosed with CHF were checked for inclusion. The final set was comprised of data from 180 CHFrEF patients who had an ischemic heart disease. The patients had not been taking any anticoagulants within the last couple of weeks, and had not experienced any factors, which could affect the platelets (PLT). All the patients gave written consent. The clinical state was considered stable if there were no changes in functional class according to the New York Heart Association (NYHA), nor changes in treatment with medicines within the past 3 to 4 weeks. Moreover, the included patients did not have any new HF symptoms. The diagnosis of CHF was performed by following the guidelines for the diagnostics and treatment of heart failure approved by the European Society of Cardiology [8]. Patients with kidney failure (eGFG < 60 mL/min.), acute or chronic infection, acute coronary syndromes, diabetes mellitus or connective tissue disease, smoking or consuming platelet-affecting agents were omitted from the study (*n* = 171). The average neutrophil count was 4.37 × 10^9^ L, therefore the patients were separated into two groups: one with relatively higher inflammatory environment (neutrophil count was ≥4.37 × 10^9^ L (*n* = 97)) and one with relatively lower inflammatory environment (neutrophil count was <4.38 ×10^9^ L (*n* = 83)).

### 2.2. Tests and Blood Sampling

The analysis of transthoracic echocardiogram, complete blood count and a 6 min walking test were performed after patients’ admission to the hospital. In order to determine and compare the levels of PLT, inflammation and stress in patients diagnosed with CHF, the PLT aggregation, the concentration of C reactive protein (CRP), N-terminal B-type natriuretic peptide (NT-proBNP), fibrinogen and cortisol were determined. Cortisol concentration was assessed at two time points: morning cortisol (cortisol_m_) at 8:00 am and evening cortisol (cortisol_e_) at 3:00 pm. Blood samples of fasting patients for the tests were taken from the forearm vein and processed as described recently [9]. All of the investigations were approved and conducted in accordance with the guidelines of the local Bioethics Committee and adhered to the principles of the Declaration of Helsinki and Title 45, U.S. Code of Federal Regulations, Part 46, Protection of Human Subjects (revised 15 January 2009, effective 14 July 2009). The study was approved by the Regional Bioethics Committee at the Lithuanian University of Health Sciences (No. BE-2-102, 20 December 2018).

### 2.3. Statistics

The statistical analysis was performed by IBM SPSS 20 (IBM Company, Chicago, IL, USA) for Windows. The statistical significance was determined at *p* < 0.05. The Levene‘s test for equality of variances was applied if the pre-test for normality (Kolmogorov–Smirnov normality test) was not rejected. Otherwise, a Mann–Whitney U test for the nonparametric data was used. Categorical variables were defined as a percentage, and comparisons were made using the chi-square test. Data are expressed as mean ± standard deviation for continuous variables and as a median (in the form Q1–Q3) for variables that proved not to be normally distributed. Pearson’s correlation test was used to assess correlation between the normally distributed data, whereas Spearman’s correlation was used to assess correlation between abnormally distributed data. The determined concentrations were expressed as a mean ± standard deviation, and the t-test was used for the comparison.

## 3. Results

The characteristics of CHFrEF patients involved in the present study are presented in Table 1. The patients (*n* = 180) were divided into two groups according to the neutrophil count. The groups did not differ in age, gender, NYHA functional class, blood pressure, left ventricular ejection fraction (LVEF), NT-proBNP and body mass index (BMI). There were no statistically significant differences in current medications usage and outcomes between the groups as well (see Table 1 and Table 2). The concentrations of CRP, fibrinogen, cortisol_m_ and cortisol_e_ were statistically significantly higher in the group with higher neutrophil counts (*p* = 0.012, *p* = 0.028, *p* = 0.029 and *p* = 0.001, respectively). Of note, neither the cholesterol concentration nor the markers of platelet function differed between the groups (Table 3). However, the PLT, leukocyte and monocyte levels were higher in the group with a higher neutrophil count (*p* = 0.032, *p* = 0.0001 and *p* = 0.0001, respectively, Table 4).

It was found that the CRP level correlated with the fibrinogen concentration (*r* = 0.408, *p* = 0.0001). Moreover, the PLT correlated with the neutrophil (*r* = 0.138, *p* = 0.041) and monocyte counts (*r* = 0.218, *p* = 0.014) and the concentration of CRP (*r* = 0.307, *p* = 0.008). The MPV data correlated with the fibrinogen concentration (*r* = 0.244, *p* = 0.004, Table 5). In addition, the WBC correlated with the concentration of fibrinogen, cortisol_m_ and CRP (*r* = 0.201, *p* = 0.008 and *r* = 0.238, *p* = 0.015, and *r* = 0.344, *p* = 0.0001, respectively), whereas the neutrophil count correlated with the concentration of fibrinogen, cortisol_e_ and CRP (*r* = 0.308, *p* = 0.0001 and *r* = 0.256, *p* = 0.009, and *r* = 0.378, *p* = 0.0001, respectively). The lymphocyte count correlated with the cortisol_m-e_ levels (*r* = 0.295, *p* = 0.003). The monocyte count was found to correlate with the concentration of fibrinogen, cortisol_m_, cortisol_m-e_ and CRP (*r* = 0.315, *p* = 0.0001, and *r* = 0.279, *p* = 0.004, and *r* = 0.228, *p* = 0.018, and *r* = 0.371, *p* = 0.0001, respectively, Table 6). Finally, the MPV correlated with the monocyte count, and reversibly correlated with the lymphocyte count (*r* = 0.317, *p* = 0.00001 and *r* = −0.188, *p* = 0.029, Table 5).

## 4. Discussion

It has been known that the neutrophil count could be linked to some unfavorable effects in patients with CHFrEF [10,11]. However, the roles of neutrophils for the development of low inflammation in advanced HF as well as the relation of neutrophils with hyperlipidemia and cortisol levels are still to be elucidated [6,7]. Therefore, we compared the levels of WBC, lymphocytes, monocytes and PLT in CHFrEF groups according to the neutrophil count. It was found that the PLT and monocyte counts were higher in the group with a higher neutrophil count and correlated with the PLT, neutrophil and monocyte counts. Moreover, the PLT correlated with the monocyte count. In addition, the MPV correlated conversely with the lymphocyte count, and correlated directly with the monocyte count. The obtained data could be better understood in light of the fact that after the myocardial infarction (MI), neutrophils infiltrate the damaged tissue and participate in the resolution of the inflammation process [6]. Neutrophils are also known to be involved in the left ventricle remodeling processes occurring in the sub-acute phase of MI [6]. Thus, certain events take place in tissues to restore homeostasis: the pro-inflammatory neutrophils are replaced by the reparative neutrophils, and the polarization of different macrophage subsets occurs [12]. It has been proposed that the changes in the levels of neutrophils within the infarcted heart tissue might be responsible for a lowered level of monocytes or deficient activation of reparative macrophages [13]. The observed events were also related to the increased levels of interleukin-4 (IL-4) and the sustained inflammatory process [13]. It is worth noting that increased neutrophil levels during the first 12 h after acute MI can signal the occurrence of CHF [14]. Moreover, activated neutrophils are known to participate in cardiac healing through polarizing monocytes and macrophages, and by clearing debris [13]. Different leukocytes, such as neutrophils, monocytes and macrophages, and their subsets take part in an inflammation-resolution program in a time-dependent manner. Thus, if neutrophils are not cleared from the infarction site or the surrounding area in time, they can contribute to the sustained inflammation [15]. It is also known that neutrophils can influence a chronic immune response [7]. Thus, our finding that fibrinogen and CRP concentrations, as well as the PLT and monocyte counts, were higher in the group with higher neutrophil levels, and the PLT and monocyte counts correlated with the concentration of fibrinogen and CRP, suggesting that (i) a higher neutrophil count could be related to a stronger inflammatory process, and (ii) in addition to the neutrophil count, the PLT and monocytes also take place in maintaining low chronic inflammation in CHFrEF patients.

The observed correlation between concentrations of acute phase proteins (CRP and fibrinogen) and blood cell count could be sustained by the established role of IL-6 in hemostasis. It has been shown that cells involved in the atherosclerotic process produce acute phase cytokines, especially IL-6, and that results in the increased production of CRP, fibrinogen, neutrophils and PLT [16]. The correlation between the CRP level and the monocyte count could be related to the changes in monocyte levels as a response to inflammation Moreover, under pathological conditions the uncontrolled inflammation can lead to the excessive release of macrophages, which can cause heart tissue damage instead of tissue healing [17].

In contrast to recent studies that proposed a positive correlation between the increased viability of neutrophils, the degree of CHFrEF and plasma CRP levels [18,19,20,21], we found a correlation only between the neutrophil count and the CRP level, and not between the neutrophil count and the severity of CHFrEF (expressed in NYHA functional class). The literature data suggest that the frequency of occurrence of complications and the degree of damage were higher in HF patients with higher neutrophil levels [4]. Thus, a possibility that both monocytes and neutrophils are involved in the inflammatory process under chronic CHFrEF cannot be neglected. In addition, we observed higher levels of PLT and monocytes in the group with higher neutrophil counts, and the correlation between the concentration of acute phase protein (CRP and fibrinogen) and the levels of PLT, neutrophil and monocyte counts. Moreover, the abovementioned data support the existence of a pro-inflammatory statement in CHFrEF patients. Therefore, an attempt to modulate the balance of monocytes and macrophages might be a promising therapeutic strategy.

Lymphopenia has also been seen to be common in HF due to the activation of the hypothalamic–pituitary–adrenal axis. The activation of this axis leads to the stimulation of cortisol secretion and finally, to the decreased lymphocyte count [22,23,24]. Interestingly, we found higher concentrations of cortisol_m_ and cortisol_e_, and lower concentrations of cortisol_m-e_ in the group with higher neutrophil counts. The obtained results indicated the relation between cortisol levels and the inflammatory environment. Thus, again, our results support the facts about inflammatory conditions in chronic CHFrEF patients.

The severity of hyperlipidemia in CVD patients (c-LDL > 3.103 mM) is known to be associated with the predominance of polymorphonuclear leukocytes, including neutrophils. This could be linked to the elevated serum levels of myeloperoxidase (MPO) and inflammatory markers, fibrinogen and CRP as well [25]. Puntoni et al. have also demonstrated that patients with familial hypercholesterolemia had higher MPO serum levels [26]. Interestingly, the MPO levels decreased with the deceased amount of total cholesterol [26]. Accordingly, a study that involved obese individuals showed that a calorie-enriched diet could increase a pro-inflammatory signaling, which was responsible for neutrophil endothelial transmigration and sustained inflammation [27]. In contrast, we did not find a statistically significant difference in any lipidogram data, i.e., the concentrations of total, LDL and HDL cholesterol and the level of triacylglycerol between the CHFrEF groups according to neutrophil count, nor a correlation between the lipidogram data and the neutrophil count. It is possible that a small sample size, heterogeneity of population and consequently the scatter of data, e.g., LDL cholesterol concentration in the group with a higher neutrophil count was 3.12 ± 1.12 mM, influenced the results. Nevertheless, there is a tendency that the higher neutrophil count corresponds to a bigger LDL cholesterol concentration (Table 3). In summary, the relationship between the cholesterol concentration and the low inflammatory condition in CHFrEF has to be further clarified.

Thus, our results suggest that the PLT, monocytes and neutrophils could be important in the maintenance of low inflammation in CHFrEF patients. Obviously, the stronger the inflammation environment is, the more stressful conditions patients have. However, the influence of hyperlipidemia on the pro-inflammatory statement is yet to be specified.

We suggest that our findings are of value because we found higher levels of CRP, fibrinogen, PLT and monocyte counts in the CHErEF group with higher neutrophil counts. In addition, we determined the correlation between the abovementioned readings and the neutrophil count. Thus, we think that the obtained results lead to several important issues. Firstly, the higher the neutrophil count is, the higher the pro-inflammatory statement patients have. This observation provides the basis for studies to identify specific neutrophil count ranges that could reflect the subclinical inflammation level. Therefore, it might be that the results of complete blood count as one of routine tests could adequately reflect the patient’s pro-inflammatory statement without additional testing for inflammatory readings, such as CRP and others. Secondly, the PLT and monocyte count levels could be also related to the maintenance of a low inflammatory environment. Therefore, it would be worth determining the reasons that sustain the pro-inflammatory condition in CHFrEF patients. Thirdly, the correlation found between neutrophil count and cortisol concentration revealed a possible relation between low inflammation and chronic fatigue. This result might provoke further analysis clarifying the way in which the patient’s immune system could be related to the inflammatory response. Finally, the statement that is usually found in the scientific literature that hyperlipidemia heightens neutrophil production, and thereby accelerates cardiovascular inflammation, should be reverified.

### Study Limitations

Our study suffers from several limitations. Firstly, the neutrophil and monocyte subsets were not evaluated. We were able to sort the patients just according to the total neutrophil count, and not according to its subsets. Thus, it would be useful to reveal a dominance of different subsets in order to clarify the development of heart failure in CHF patients. Secondly, the target population size was not sufficiently large. The bigger population would have increased the power of research. In addition, there was no control group. Thirdly, the concentrations of other, more precise inflammatory markers, for example, IL-6, IL-4 or TNF-α, were not evaluated. The following data would be very helpful for a deeper understanding of the relationship between the obtained data, the inflammatory reaction and the CHF degree. Information about the concentration of lipoprotein subclasses could have also been valuable in order to reveal a relationship between cholesterol concentration and low inflammation in CHFrEF patients.

We would like to emphasize that neither correlation nor a simple comparison revealed any causal relationship between the investigated readings. Our findings are important for supplementing the knowledge about inflammatory, prothrombotic, lipidemic and stressful conditions in CHFrEF patients. Thus, to the extent of our knowledge, this is the first work presenting the differences in the levels of monocytes, lymphocytes and platelets, and also the differences in the concentrations of lipids and cortisol between the two groups of CHFrEF patients with different pro-inflammatory statements (according to the neutrophil count).

## 5. Conclusions

We determined that the concentrations of inflammatory markers, such as CRP and fibrinogen, were higher in the group with higher neutrophil counts. This suggests that the higher neutrophil count is, the higher the inflammatory condition CHFrEF patients could experience. Morning and evening cortisol concentrations were higher in the group with higher neutrophil counts. The obtained correlations between the cortisol concentration and the neutrophil and monocyte counts suggest a possible relation between stress and low inflammatory statement in CHFrEF patients. The PLT and monocyte counts were higher in patients with the higher neutrophil counts as well. However, the platelet aggregation and MPV did not differ between the groups. Therefore, it could be assumed that the PLT and monocytes, together with neutrophils, sustain the pro-inflammatory environment. Cholesterol levels and platelet function readings did not differ between the groups.

In general, our findings support the knowledge about subclinical inflammation in CHFrEF patients. However, more precise inflammation data, neutrophil subsets and concentration values of the lipoprotein subclasses are needed to better understand the importance and relation between low inflammation and cholesterol concentration in CHFrEF patients.

## Figures and Tables

**Table 1 medicina-57-00557-t001:** Patient characteristics in groups according to the neutrophil count.

Characteristics	≤4.37 × 10^9^ L *n* = 97	>4.38 × 10^9^ L *n* = 83	*p*
Age	56.68 ± 14.12	52.94 ± 13.73	0.215
Female	23 (23.7%)	14 (16.9%)	0.274 *
NYHA I	15 (16.9%)	11 (15.3%)	0.830 *
NYHA II	32 (36%)	27 (37.5%)	0.640 *
NYHA III	30 (33.7%)	21 (29.2%)	0.730 *
NYHA IV	12 (13.5%)	13 (18.1%)	0.890 *
SBP mmHg	125.97 ± 17.57	128.25 ± 22.57	0.698
DBP mmHg	82.34 ± 11.57	80.49 ± 14.25	0.745
BMI	27.24 ± 4.44	28.32 ± 5.63	0.597
LVEF, %	31.66 ± 11.65	30.68 ± 12.74	0.197
NT-proBNP, pg/L	530.60 (29–8302.0)	749.30 (50.6–15,377.0)	0.765
Atrial fibrilation	13 (40.6%)	13 (44.1%)	0.562 *
Thrombosis	5 (15.6%)	3 (11.5%)	0.654 *

BMI—body mass index, SBP—systolic blood pressure, DBP—diastolic blood pressure, LVEF—left ventricular ejection fraction, NT-proBNP—N terminal B-type natriuretic peptide; * the comparisons were made using the chi-square test; the Student t-test was used for the variables showing the parametric distribution; the Mann–Whitney U test was used for the nonparametric data.

**Table 2 medicina-57-00557-t002:** Drug usage in the groups according to the neutrophil count.

Drug	≤4.37 × 10^9^ L *n* = 97	>4.38 × 10^9^ L *n* = 83	*p*
ACE-inhibitors (*n*)	45 (46.4%)	37 (44.6%)	0.808
Heparine (*n*)	2 (2.1%)	5 (6.0%)	0.170
Diuretics (*n*)	29 (29.9%)	21 (25.3%)	0.493
Beta-blockers (*n*)	50 (51.5%)	34 (41.0%)	0.156
Nitrates (*n*)	3 (3.1%)	8 (9.6%)	0.068
Digoxin (*n*)	8 (8.2%)	7 (8.5%)	0.945
Statins (*n*)	5 (5.2%)	1 (1.2%)	0.141
Calcium channel blockers (*n*)	3 (3.1%)	2 (2.4%)	0.781

The comparisons were made using the chi-square test.

**Table 3 medicina-57-00557-t003:** Laboratory data in groups according to the neutrophil count.

Levels	≤4.37 × 10^9^ L *n* = 97	>4.38 × 10^9^ L *n* = 83	*p*
Fibrinogen concentration, g/L	4.00 (2.01–6.11)	4.42 (2.35–7.69)	0.028 *
CRP, mg/L	3.10 (1.00–26.76)	4.9 (1.00–90.50)	0.017 *
Total cholesterol, mM	4.72 ± 1.10	4.74 ± 1.49	0.332
HDL cholesterol, mM	1.19 ± 0.37	1.13 ± 0.36	0.660
LDL cholesterol, mM	2.89 ± 0.78	3.12 ± 1.12	0.189
Triglycerides, mM	1.31 ± 0.58	1.29 ± 0.79	0.192
ADP, %	61.94 ± 19.74	62.56 ± 19.93	0.992
ADR, %	66.88 ± 25.10	64.95 ± 27.75	0.125
Cortisol_m_ mM	460.94 ± 155.59	491.67 ± 150.84	0.029
Cortisol_e_ mM	350.19 ± 95.78	409.14 ± 114.92	0.007
Cortisol_m-e_ mM	110.75 ± 125.68	82.53 ± 132.88	0.208

CRP—C reactive protein, HDL—high density lipoprotein, LDL—low density lipoprotein, ADP—adenosine diphosphate, ADR—epinephrine, Cortisol_m_—morning cortisol concentration, Cortisol_e_—evening cortisol concentration, Cortisol_m-e_—difference of morning and evening cortisol concentrations; * the comparisons were performed using the Mann–Whitney U test; the Student t-test was used for the variables showing the parametric distribution.

**Table 4 medicina-57-00557-t004:** Complete blood count within groups according to the neutrophil count.

Levels	≤4.37 × 10^9^ L *n* = 97	>4.38 × 10^9^ L *n* = 83	*p*
PLT × 10^9^/L	209.45 ± 42.75	238.07 ± 89.87	0.032
MPV, Fl	10.35 (7.5–12.6)	9.62 (7.7–12.8)	0.536 *
WBC × 10^9^/L	5.81 ± 1.15	8.51 ± 1.68	0.0001
Lymphocytes × 10^9^/L	1.71 ± 0.68	1.89 ± 0.83	0.109
Monocytes × 10^9^/L	0.56 ± 0.21	0.72 ± 0.27	0.0001

PLT—platelet count, MPV—mean platelet volume, WBC—leukocyte count; * the comparison was performed using the Mann–Whitney U test; the Student t-test was used for the variables showing the parametric distribution.

**Table 5 medicina-57-00557-t005:** Correlation between blood cell counts.

Levels/*r*,*p*	WBC	Neutrophil Count	Limphocytes Count	Monocytes Count
PLT	0.306, 0.0001	0.183, 0.041		0.218, 0.014
MPV			−0.188, 0.029	0.317, 0.0001

PLT—platelet count, MPV—mean platelet volume, WBC—leucocyte count.

**Table 6 medicina-57-00557-t006:** Correlations between blood cell count and laboratory data.

Levels/*r*,*p*	PLT	MPV	WBC	NEU	LIMFO	MONO
Fibrinogen, g/L	0.180, 0.042	0.244, 0.004	0.201, 0.008	0.308, 0.0001		0.315, 0.0001
Cortisol_m_, mM			0.238, 0.015			0.279, 0.004
Cortisol_e_, mM				0.256, 0.009		
Cortisol_m-e_, mM					0.295, 0.003	0.228, 0.018
CRP, mg/L	0.307, 0.008		0.344, 0.0001	0.379, 0.0001		0.371, 0.0001

Cortisol_m_—morning cortisol concentration, Cortisol_e_—evening cortisol concentration, Cortisol_m-e_—difference of morning and evening cortisol concentrations, PLT—platelet count, MPV—mean platelet volume, WBC—leucocyte count, NEU—neutrophil count, Limpho—lymphocyte count, Mono—monocyte count.

## Data Availability

The study did not report any data.

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
