# Peer review of "The Difference of Cholesterol, Platelet and Cortisol Levels in Patients Diagnosed with Chronic Heart Failure with Reduced Ejection Fraction Groups According to Neutrophil Count"

_medicina, 2021, doi:10.3390/medicina57060557_

Round 1

Reviewer 1 Report

The authors did some improvements to the manuscript. Maybe a good idea would be to mention in the Material and Methods section that data is expressed as: as mean ± standard deviation for continuous variables and as median (in the form Q1-Q3) for variables that proved not to be normally distributed.

Author Response

The comment in the Material and Methods section that data is expressed as: as mean ± standard deviation for continuous variables and as median (in the form Q1-Q3) for variables that proved not to be normally distributed have been added.

Reviewer 2 Report

The authors modified the text according my previous questions. I have no other comments. 

Author Response

Thank you very much for your review

This manuscript is a resubmission of an earlier submission. The following is a list of the peer review reports and author responses from that submission.

Round 1

Reviewer 1 Report

This study was aimed to compare the complete blood count parameters (lymphocytes, monocytes, platelets count), platelet characteristics (mean platelet volume, platelet aggregation), cortisol levels, lipidogram and inflammation markers (CRP, fibrinogen concentration) in patients with chronic heart failure.

First of all, English revision is needed to improve the clarity of the manuscript, as some of the phrases are not clear enough. For example: “It seems could be useful to evaluate not only lipidogram” (line 271) is not clear what the authors wanted to say, “Secondly, the investigated population was not big.” (line 283), maybe here the authors intended to stress that the target population size was not sufficiently large.

All abbreviations should be defined at their first appearance in the text, even in the manuscript.

What the authors wanted to say with “[reviewed in 3]” (line 53) or “[reviewed in 6]” (line 60)

The Abstract should be reorganized since too many p and r values are given.

Table 1. from Methods with patients’ characteristics in groups according to neutrophil count should be moved to results.

Under Table 3 and Table 4 information about how the data is expressed should be given.

Tables 5, 6, and 7 should also provide information about the data expressed in the tables.

Authors should define MPO abbreviation (line 258)

Author Response

Thanks to the Reviewer for carefully reviewing the article and helpful comments. We have corrected the article in the light of the comments. It improved the quality of the article. The following corrections have been made:

1.English revision have been done.

2.The phrase “It seems could be useful to evaluate not only lipidogram” has been cleared and changed to “It seems could be useful to evaluate not only lipidogram (it involves total cholesterol concentration, low density and high density lipoprotein cholesterol concentration and triacylglycerols concentration), but the concentration of a different lipoprotein subclasses (they involves low density and high density lipoprotein subclasses’ number) too”.

3.The phrase “Secondly, the investigated population was not big.” has been changed to “the target population size was not sufficiently large”.

  1. All abbreviations have been defined at their first appearance in the abstract and in the manuscript.
  2. “[reviewed in 3]” (line 53) and “[reviewed in 6]” (line 60) have been corrected in to [3] and [6]. Previous indices remained in the text by mistake.
  3. The Abstract have been reorganized removing least important p and r values.
  4. Table 1. from the Methods with patients’ characteristics in groups according to neutrophil count have moved to results.
  5. Under Table 3 and Table 4 information about how the data is expressed have been given: 3 table: *- comparisom was made using Mann-Whitney test. t test was used for others.. 4 table: *- Mann-Whitney test. T test for other comparison was applied. In first column of 3 and 4 tables are given all units in what readings are expressed.
  6. The MPO abbreviation heve been defined (line 258).
  7. Information have been provide about the data expressed in the Tables 5, 6. Table 5 have been removed.

Reviewer 2 Report

This is a study about the relation between neutrophils and other inflammatory markers in patients with HFrEF. Methods description is quite confusing as well as results and thus, even if the discussion is quite extensive, the final message of the studies is difficult to be it is hard to get caught. I suggest an extensive revision of the grammar of the study, reducing the amount of text, focusing the attention of the reader on the real message of the study.

Please revise methods of the abstract because at present are really confusing.

Why the cut-off of 4.37 x 103 have been used for the stratification of the two groups? Please explain it in abstract.

Please change “cortizol” with “cortisol” in title

What does it mean [reviewed in 3] or  [reviewed in 6]? Please amend.

Please change the world “reading” with “levels”.

Please add the world “admitted to the Department of Cardiology  between 01.01.2016-01.03.2018”.

Please change “heart echoscopy” with “transthoracic echocardiogram”

Was blood cortisol concentration assessed in fasting patients?

In statistics is written twice that t test was used for comparison of normally distributed variables expressed by means.

Please add in results the neutrophil count according NYHA IV.

Please do not use the term “platelet aggregation reading” is better something like “other markers of platelet function”.

Which statistic text was used to assess correlation?

Why to present results of correlation of variables like CRP, SBP, MPV, PLT etc.. Is it not better to focus the attention of the reader on neutrophil? Which is the main endpoint of the study? Please better explain it in  methods.

As limitation of the study it has to be noted that there is not a control group.

Author Response

Thanks to the Reviewer for carefully reviewing the article and helpful comments. We have corrected the article in the light of the comments. It improved the quality of the article. The following corrections have been made:

  1. Methods of the abstract have been revised and following changes have been made: additional abbrevations have been defined, the sentence about patients have been shortened.
  2. The explanation, why the cut-off of 4.37 x 103have been used for the stratification of the two groups have been added in the abstract.
  3. “Cortizol” with “cortisol” in title have been changed.
  4. “[reviewed in 3]” (line 53) and “[reviewed in 6]” (line 60) have been amended in to [3] and [6].
  5. The world “admitted to the Department of Cardiology  between 01.01.2016-01.03.2018” have been added.
  6. “heart echoscopy” with “transthoracic echocardiogram” have been changed.
  7. Yes, blood cortisol concentration assessed in fasting patients. It have been added in the Methods.
  8. Statistics part have been corrected.
  9. We choosed to remove the sentence about neutrophil count between the groups according NYHA, because the paper’s aim was to compare results according to neutrophil count. So sentence about groups according to NYHA was misleading.
  10. The term “platelet aggregation reading” have been replaced in “ markers of platelet function”.
  11. Pearson’s correlation test was used to asses correlations between the normal distribution data, and Spearman’s correlation was used to asses correlations between the abnormal distribution data. This explanation have been added in Statistics part.
  12. Results of correlation of SBP are removed from the text as excess. Table 5 have been removed too.
  13. The main endpoint of the study was to find out a relations between platelet, inflammation and stress readings in chronic heart failure with reduced ejection fraction patients. It have been explained in Abstract and Introduction.
  14. As limitation of the study it have been noted that there is not a control group.